# An Overview of Achilles Tendinopathy Management

Fabio Santacaterina, Sandra Miccinilli, Federica Bressi, Silvia Sterzi and Marco Bravi *

Department of Physical and Rehabilitation Medicine, Università Campus Bio-Medico di Roma, Via Alvaro del Portillo, 200, 00128 Rome, Italy; f.santacaterina@unicampus.it (F.S.); s.miccinilli@unicampus.it (S.M.); f.bressi@unicampus.it (F.B.); s.sterzi@unicampus.it (S.S.)
* Correspondence: m.bravi@unicampus.it; Tel.: +39-06-22541-624

**Abstract:** Background: Persistent tendon pain and swelling related to mechanical loading are the main signs of Achilles tendinopathy (AT). This condition is one of the most common tendinopathies of the lower limb affecting mainly athletes involved in running and jumping sports. Methods: we included pivotal papers retrieved from the literature (Pubmed, Google Scholar, PEDro, and Scopus) to present an overview of the management of AT, with a specific focus on conservative management. Results: An accurate and timely diagnosis of AT is necessary to set up early treatments and to manage the problem conservatively. Diagnosis is primarily based on clinical assessment; instrumental imaging may be helpful in confirming the clinical diagnosis. Conservative treatment is effective in most cases, mainly using physical exercise based on eccentric training. Other non-surgical treatments such as extracorporeal shock wave therapy, thermotherapies, and injections can be added to exercise. Surgical treatment is indicated for patients where the conservative treatments of at least six months fails. Conclusions: Conflicting results from numerous studies hamper to identify gold standard treatments asking for further well-conducted level I and II research about the management of AT.

**Keywords:** Achilles tendinopathy; tendon; eccentric training; injections; rehabilitation; overview; athletic injury; manual therapy; physical therapy; conservative treatment; physical exercise





## 1. Introduction

Achilles tendinopathy (AT) is a clinical condition characterized by persistent tendon pain and swelling related to mechanical loading of the Achilles tendon [1–3]. AT, along with plantar fasciitis and stress fractures, are the injuries that most frequently involve the foot [4], furthermore AT is one of the most common tendinopathies of the lower limb [5]. The incidence rate of AT is 1.85 per 1000 patients visiting general practitioners, rising up to 2.35 per 1000 in the population aged 21–60 years [6]. Among others, the population most exposed to develop AT are athletes, especially those who practice running and jumping sports. In fact, de Jonge et al. reported a relationship with sports activity in 35% of AT cases [6]. Moreover, a recent study by Janssen et al. [5] reported that among elite track and field athletes, 43% reported having symptoms of AT, with the highest prevalence of AT found in medium- and long-distance runners. Furthermore, the perception of stiffness in the calf muscles appears to be a risk factor associated with the development of AT [5]. However, AT also occurs in non-athletes population especially among middle-aged overweight patients who have not faced increased physical activity [7].

The identification of risk factors is still unclear. the study by Holmes et al. reported that individual patient characteristics such as age, male gender, and obesity have been shown to have positive correlation with AT [8]. On the contrary, the study by Longo et al. on a population of track and field athletes, found that although most of the studies conducted show a prevalence of male gender, does not seem to be a correlation between age, gender, weight, height, and impact profile in the development of AT [9]. Chronic AT is thought to be the result of repetitive overuse injuries, with a tenfold increase in Achilles

tendon injuries in runners compared to age-matched controls [10,11]; however, even in patients without active participation in strenuous physical activity, AT often occurs [12].

Intrinsic and extrinsic factors would seem to represent causal factors of AT although the debate regarding the etiology remains open. Tendon vascularity as well as the weakness and lack of flexibility of the gastrocnemius and soleus muscles, the presence of pes cavus and lateral ankle instability are the so-called intrinsic factors [13]. The main risk factor for developing AT is excessive load on the tendon [14]. Another etiological factor of AT, reported by Maffulli et al. [15], is represented by the damage of free radicals which occur after ischemia, hypoxia, hyperthermia, and reduced apoptosis of the tenocytes.

Staff involved in the management of AT face a serious challenge. In fact, the results even after surgery are different and the surgery itself requires prolonged rehabilitation [15]. Moreover, symptoms can last between 3 to 12 months after starting treatment, in about 25% of patients chronic tendinopathy related symptoms can be present even after 10 years [16].

Given the high incidence of AT, the purpose of this work is to help the reader with an overview of the approach to the patient with AT, from diagnosis to the various therapeutic opportunities currently used and reported in the literature. We present an overview of the management, from diagnosis to treatment, of AT, with a specific focus on conservative management based on pivotal works retrieved from the following database: Pubmed, Google Scholar, PEDro, and Scopus.

## 2. Diagnosis

One of the key factors for successful management of AT is the early definition of the condition, therefore the diagnosis of AT must be accurate and timely. The identification of clear diagnostic criteria helps to identify the problem, set the appropriate treatment, and determine the prognosis [17]. Although to date there are several medical diagnostic tests (e.g., clinical examination, imaging, clinical tests; see Table 1), it is not easy to define the exact diagnostic tests and symptoms to diagnose AT [17].

**Table 1.** AT Diagnosis: physical examination and imaging.

| Physical Examination | Imaging |
| --- | --- |
| Royal London Hospital Test (RLHT)<br>Sensitivity 54%<br>Specificity 86% | Radiographs<br>(symptoms lasting over 6 weeks) |
| Palpation<br>Sensitivity 64%<br>Specificity 81% | Ultrasound<br>(to confirm clinical diagnosis) |
| The Arc sign<br>Sensitivity 42%<br>Specificity 82% | Magnetic resonance imaging<br>(used for preoperative planning) |

The diagnosis of AT is mainly based on the medical history and physical examination [18]. One of the initial symptoms of AT is morning stiffness or stiffness after inactivity, while pain is a late symptom [15]. Nevertheless, the type of pain is a key feature in identifying AT: initially the pain appears as a pain that may not be disabling, but with continued physical activity it can affect training skills. Pain usually decreases with rest but exacerbates with physical activity. Furthermore, according to Cook et al. [19] morning pain is a hallmark of AT and along with stiffness are considered good indicators of tendon health. A recent work by de Vos et al. [17] showed that experts generally agree in diagnosing AT when localized tendon pain, tendon thickening, and pain associated with weight-bearing activities are present. Additionally the presence of pain on palpation localized in the distal 2 cm of the Achilles tendon associated with swelling and redness of the area should lead to suspect AT [20].

Physical examination plays a pivotal role for the diagnosis of AT, as extensively discussed by Maffulli et al. [15], as it allows excluding other conditions that can cause

similar symptoms. A main body tendinopathy of the Achilles tendon presents, on physical inspection, a tender area of intratendinous swelling that moves with the tendon and whose tenderness diminishes or noticeably disappears when the tendon is stretched. An insertional AT shows pain which generally arise from the heel and is exacerbated by active and passive mobilization [15].

There are three main clinical measures consisting of simple maneuvers during physical examination, which can help clinicians in diagnosing AT:

1.  Royal London Hospital Test (RLHT): the test is performed with the patient lying in the prone position with the foot off the edge of the bench and the ankle in a neutral position. The test consists of palpating the Achilles tendon to search for tenderness. Then, the tenderness spot is palpated in maximum dorsiflexion of the ankle and in maximum plantar flexion of the ankle (Figure 1). The test is considered positive for AT if the pain on the initially identified tenderness spot is absent in maximum ankle dorsiflexion [21]. A meta-analysis by Reiman et al. found a pooled sensitivity of 54% and a specificity of 86%, but the test has moderate clinical value due to the risk of bias of the included studies [22].
2.  Palpation: this maneuver consists of a gentle palpation of the whole tendon length squeezing the tendon, proximally to distally, between the thumb and the index finger. The test is considered positive if the patient reports pain [23]. The palpation has a pooled sensitivity of 64% and a specificity of 81% [22].
3.  The Arc sign: the test is performed with the patient in the same position as the RLHT. The clinician palpates the tendon in a distal to proximal direction searching for localized thickening of the tendon. Then holding the fingers on the area of swelling the patient is asked to perform an ankle dorsiflexion and plantarflexion. The test is considered positive if the area of swelling moves with the ankle movement. The Arc sign test has a pooled sensitivity of 42% and a specificity of 88% [22].

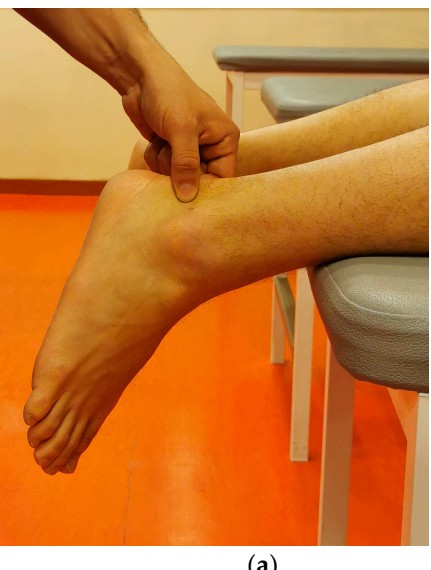 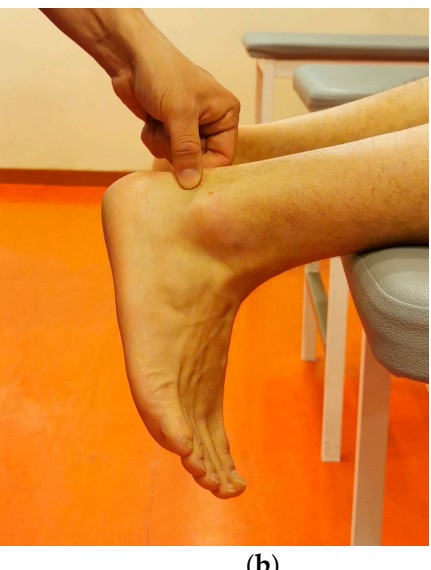

(**a**) (**b**)

**Figure 1.** Royal London Hospital Test: First the clinician palpates the tendons to search tenderness (**a**). Then the patient is asked to actively dorsiflex the ankle and to actively plantarflex it. The test is considered positive for AT if the pain on the initially identified tenderness spot (**a**) is absent in maximum ankle dorsiflexion (**b**).

Subjective self-reported pain with a localization of pain 2–6 cm above the insertion of the Achilles tendon to the calcaneum, and subjective reporting of morning stiffness with a pain that is usually worse in the morning, are subjective assessments that a clinician must consider when there is a suspicion of AT. Although Hutchison et al. [21] found that

the tests, which demonstrated the highest validity, were "self-reported pain" and "pain on palpation".

The severity of AT can be measure through the Victorian Institute of Sport Assessment-Achilles (VISA-A). VISA-A explores the domains of pain, function, and activity. The score of 100, obtained from the sum of the scores of the individual domains, corresponds to the value that a healthy person would obtain. The VISA-A questionnaire provides a valid, reliable and easy to use index to assess the severity of AT [24,25].

Imaging offers three methods for diagnosing AT: radiographs, ultrasound (US) and magnetic resonance imaging (MRI).

Radiographs are usually required for patients presenting symptoms lasting more than six weeks; generally, radiographs of the foot can be used to evaluate the presence of a posterior heel spur (sign of insertional AT), size of enthesophytes, intratendinous calcifications, and a Haglund deformity [15,20]. Furthermore, a retrocalcaneal bursitis can be diagnosed by analyzing the shape and brightness of Kager triangle on radiographic imaging [26].

US is usually used to confirm the clinical diagnosis of AT through a visualization of tendon structure; however it has a poor ability to detect an early tendon damage and to monitor the progress of the pathology or healing. Furthermore, there seems to be a poor correlation between imaging and clinical symptoms [27]. Therefore, an innovative computerized imaging modality called "ultrasound tissue characterization" (UTC) was developed by van Schie et al. [28], UTC allow quantitatively assessing the tendon structure and to discriminate symptomatic from asymptomatic tendons [28,29].

MRI provides additional information on the morphology of the tendon and surrounding bone and allows distinguishing between healthy and diseased tendons. It allows the surgeon to estimate the extent of diseased tissue present for preoperative planning [15]. However, the ability to predict symptoms is limited, in fact there would seem to be little correlation between the abnormalities identified by MRI and the symptoms reported by patients [30]. Furthermore, the ability of MRI to monitor response to treatment and progress of healing is still debated. A recent study by Tsehaie et al. showed that MRI parameters add no value in daily clinical practice in providing a prognosis [31].

## 3. Treatment

### 3.1. Conservative Treatment

Several conservative approaches are available for the treatment of AT. In general, the conservative approach represents the first therapeutic choice for the treatment of TA and should last for three to six months before considering surgery, in fact during this time the AT tends to resolve in 75% of patients [15].

### 3.1.1. Exercise Therapy

Exercise rehabilitation provides the best level of evidence in tackling AT [1,32–34]. Different types of exercise are used in the conservative management of AT (Table 2). All the exercises are based on the optimal load to the tendon in order to guarantee pain reduction, improve lower limb muscles strength and flexibility, promote remodeling and leg function. Factors included in the optimal loading have not yet been defined but loading programs seem to be a good approach strategy [35–37]. The loading approach involves the patients in a graduated program, using progressions guided by time-based criteria and symptom severity [38]. In general, it appears that tendons have a better response to high loads with longer duration than undergoing low loads with reduced duration [39].

**Table 2.** Type of exercise for AT.

| Type of Exercise | Study | Sets & Reps | Frequency | Type of Load |
|---|---|---|---|---|
| Eccentric | Alfredson et al., (1998) | 3 sets, 15 reps | Twice daily for 12 weeks | Bodyweight initially. Increased as pain allows |
| Combined | Silbernagel et al., (2007) | Various combination of sets & reps | Daily for 12 weeks to 6 months | Bodyweight initially. Increased in phases based on patient status |
| Heavy slow resistance training | Beyer et al., (2015) | 4 sets, 15 to 6 reps | 3 times per week for 12 weeks | The number of repetitions decreased, and load gradually increased, every week as the tendon got stronger. 3 times, 15-repetition maximum (15 RM), in week 1; 3 times, 12 RM, in weeks 2 to 3; 4 times, 10 RM, in weeks 4 to 5; 4 times, 8 RM, in weeks 6 to 8; and 4 times, 6 RM, in weeks 9 to 12. |

Isometric exercise was recently proposed in the early stages of AT treatment [40], but there is no evidence of its superiority over any other exercise modality [41] and does not seem to have immediate effects in reducing pain in patients with chronic AT [42].

To date, eccentric loading exercise programs are considered the principal approach to AT [43], which was confirmed by recent studies that demonstrate a strong evidence in eccentric training exercises (Figure 2) [44,45]. In recent decades, the Alfredson's eccentric exercise protocol published in 1998 was long considered the best exercise protocol for AT conservative treatment. This program consists of two different eccentric exercises (3 × 15 repetitions 2 times daily, 7 days/week, for 12 weeks) asking the patient to perform heel drops on the injured ankle for loading eccentrically the plantar flexor muscle-tendon unit while using the healthy lower limb to return to the start position [46]. Sayana & Mafulli [47] showed that despite the large use of this protocol, 45% of individuals did not responded favorably to the protocol, thus questioning its wide popularity; for this reason Jonsson et al. [48] proposed a modified version of the Alfredson's protocol without loading into dorsiflexion to improve efficacy in patients with AT.

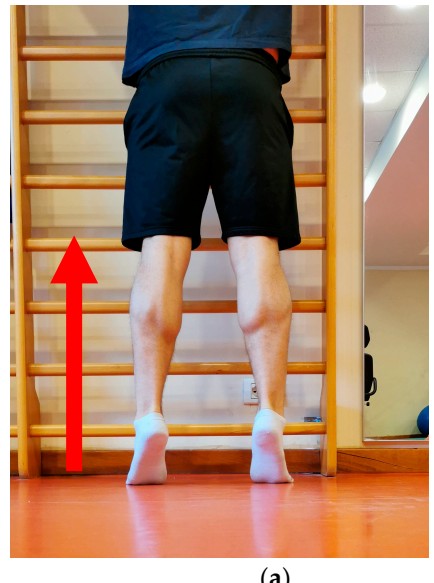 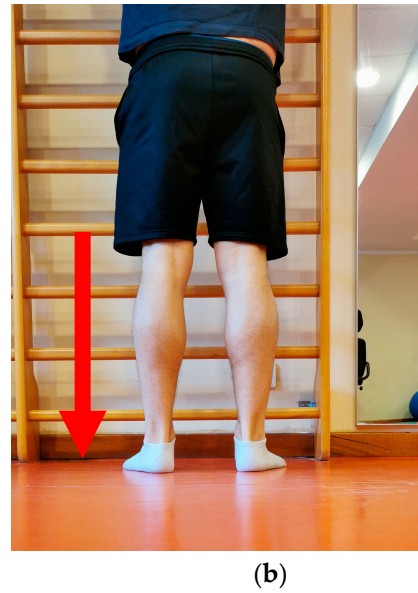

(**a**)  (**b**)

**Figure 2.** Eccentricexercise for AT. Raise both heels and put weight over big toes (concentric contraction) (**a**). Slowly lower heel down to starting position (eccentric contraction) (**b**).

Past studies demonstrated how the response of collagen to loading is quite slow [49,50], for this reason the loading attitude of tendons could be impaired for 2 days after exercise. Therefore, a recovery period of 2 days between the exercises sessions is recommended as suggested by Silbernagel and Crossley [38].

In 2015 Beyer et al. demonstrated that heavy slow resistance training (HSRT), with a loading frequency of three times weekly, produces good results with greater patient satisfaction after 12 weeks if compared to Alfredson's program [43]. A recent systematic review suggests that patients with AT could benefit more from a lower loading frequency programmes compared to Alfredson's high loading frequency program [51]. Silbernagel et al. [1] in 2020 found that exercises providing a progressive tendon loading promote the remodeling and restoration of calf-muscle function. The method shows the highest evidence.

### 3.1.2. Extracorporeal Shockwave Therapy

Extracorporeal shockwave therapy (ESWT) as a treatment approach for AT involves the delivery of shock waves directly to the painful area of the tendon. ESWT is a safe and well tolerated treatment modality for AT [52]. Focused ESWT and radial ESWT are the two different ESWT delivery modality available. Focused ESWT has greater penetration power and impact force than radial ESWT, which is directed radially to the skin [53]. A recent study by Santamato et al. [53] showed that patients with non-insertional AT treated with focused ESWT achieved significant pain reduction and functional improvement at 1 and 3 months of follow-up. However, the authors found no efficacy in developing neovascularization. To date it is not possible to establish which is the best treatment modality with ESWT in fact there are no studies that compare the effects of radial shock wave therapies and focused on AT [52].

Currently there is still a lot of variability in ESWT treatment parameters in terms of energy level (expressed in $mJ/mm^2$) which is determined by number of impulses per treatment session and pressure (expressed in bar), frequency of impulses, number of weekly sessions and number of total sessions. This makes it difficult to identify a standard way of treating AT with ESWT. A recent meta-analysis conducted by Fan et al. [54] found that medium energy ESWT (0.12–0.25 $mJ/mm^2$) was comparable to low energy ESWT (0.06–0.11 $mJ/mm^2$) in improving pain VAS scores; furthermore, they found that ESWT led to better VAS pain scores than other conservative treatments at both shorter (<6 months) and longer (>6 months) follow-up. However, a recent work, not included in the meta-analysis by Fan et al., compared the use of two different applicators, point-focused and the line-focused applicator, in the treatment of AT. The authors found significant VISA-A score improvement for all groups (including placebo group) during 24 weeks without a statistically significant outcome difference between groups [55].

### 3.1.3. Other Physical and Manual Therapies

The use of low-level laser therapy (LLLT) for the treatment of soft tissue injuries and inflammation began in the 1960s. Wavelengths between 660 nm and 905 nm are able to penetrate the skin and soft/hard tissues, causing positive effects on pain, inflammation and tissue repair [56]. Furthermore, it has been shown that LLLT could increase the synthesis of collagen [57] and angiogenesis [58]. A meta-analysis by Tumilty et al. [59] in 2010 indicated that LLLT could potentially be effective in treating tendinopathy when recommended dosages were used, but the heterogeneity of the studies hampered to establish the exact efficacy of LLLT. A more recent meta-analysis focused on the use of LLLT in AT showed low to very low evidence highlighting the lack of sufficient data to support the clinical effects of LLLT for AT [60].

Although ultrasound therapy (UT) is widely used in rehabilitation, there is insufficient evidence to support a clinically beneficial effect. As reported by Warden et al. [61] UT could have a beneficial effect on tissues under an acute inflammatory reaction. To date, few studies have analyzed the effects of UT on tissues such as ligaments, tendons and muscles;

therefore it is not possible to establish their real clinical efficacy in the treatment of AT [61]. A randomized and controlled pilot study showed safety in the use of UT, but no superiority of efficacy over eccentric exercises in the treatment of sedentary subjects with AT [62].

Heat therapy, which induces a rise in temperature on the application area, can be an effective method of treating tendon disorders [63]. The use of heat in the management of AT lead to an improvement of blood flow and oxygen saturation in the Achilles tendon [64]. The use of thermotherapies such as hyperthermia and capacitive-resistive electric transfer would appear to be effective in the treatment of AT [64–66].

The use of cryotherapy is debated, it seems to be effective in increasing tendon oxygenation [67] and improving tendon thickness [65], but there is insufficient evidence to prove its efficacy as a treatment for AT. A recent work by Dubois et al. [66] which presents a new acronym guiding management PEACE and LOVE for soft-tissue injuries, has raised doubts about the use of ice indeed there is no high-quality evidence to support the effectiveness of ice. Moreover, ice could potentially interrupt the inflammatory process angiogenesis and revascularization which could cause a reduction in the repair of damaged tissues. Deep friction massage (DFM) is a technique widely used by physiotherapists in the treatment of tendinopathies [68]. As far as we know there are no studies on the efficacy of DFM for the treatment of AT. A review by Joseph et al. on the effectiveness of DFM in the management of tendinopathies reports that it helps to restore tissue elasticity and to reduce strain in the muscle-tendon unit in combination with other treatments [69]. However, there is insufficient evidence to define the real effectiveness of DFM in fact the same review demanded for further high-level studies to define DFM effectiveness.

### 3.1.4. Injections

Injections (a further conservative treatment of AT) make use of numerous substances as extensively reported by Maffulli et al. [15]. High volume injections are popular [70]. Recently the scientific community is orienting research towards autologous blood-derived products (ABP). Some studies have shown that the use of platelet-rich plasma (PRP) helps to obtain a better tendon healing [71,72], for this reason the use of PRP injections has become widely used in the orthopedic field. However, to date the results are mixed, Boesen et al. [73] reported that PRP injection associated with eccentric training is effective in conservative management of chronic AT while de Jonge et al. in their RCT involving 45 patients showed that the use of PRP injections associated with an eccentric training program did not show clinical superiority over placebo saline injections [74]. Furthermore, recent meta-analysis showed that the use of ABP (including PRP) does not add beneficial effects thus preventing their recommendation for use in clinical practice [75–77]. A recent RCT by Usuelli et al. [78], whose purpose was to compare the effectiveness of a PRP injection with an adipose-derived stromal vascular fraction (SVF) injection for the treatment of AT, found that both PRP and SVF injection were safe and effective in the treatment of AT. Furthermore, they found that patients treated with SVF obtained faster results suggesting its use in those patients who need an early return to sports activity.

### 3.2. Surgical Treatment

If a benefit is not obtained after six months of conservative treatment (which can happen to 24–45.5% of patients with AT) [15,79], surgical treatment can be used. There are three main surgical options available: simple percutaneous tenotomy, minimally invasive stripping of the tendon and open procedures [80]. Worst results are expected in women after surgical treatment [81].

The simple percutaneous tenotomy consists of an area of variable tenolysis, about 4 cm long, obtained through an incision. The procedure involves the repetition of additional four incisions (2 cm medially and proximally, medially and distally, laterally and proximally and laterally and distally) [82].

The minimally invasive stripping described by Longo et al. [83] consists of stripping and freeding of the proximal and distal portions of Achilles tendon of all the peritendi-

nous adhesions. The procedure can involve both the anterior and posterior side of the Achilles tendon.

The open procedures include generally longitudinal tenotomies with debridement of the tendon. This procedure may be performed with or without tendon augmentation (a tendon augmentation or transfer should be considered if more than 50% of the tendon is debrided [15]), gastrocnemius lengthening or recession.

A recent systematic review by Lohrer et al. [84] included twenty studies comparing minimally invasive procedures to open procedures and reported a mean success rate of surgery of 83.4%. The minimally invasive procedures reported an average success rate of 83.6% while the open procedures 78.9%. They also observed a tendency for an increase of complications in open procedures (complication rate: 5.3% for the minimally invasive techniques, 10.5% for the open procedures, $p = 0.053$).

Post-operative rehabilitation involves an initial healing phase whose focus is to early mobilize the ankle avoiding the functional overload of the tendon. In the immediate post-operative period, generally for the first 14 days, the patient is asked to use a splint and to walk with crutches. Subsequently, passive and active mobilization exercises of the ankle, to be performed every day, are allowed and prescribed [15].

The decision about weight-bearing the operated side is based on the type of surgery and the degree of debridement performed during the surgery. Generally, as reported in the review by Lohrer et al. [84] after open procedures, weight-bearing is usually allowed after 2 weeks, and the use of a cast is recommended for over 6 weeks, while minimally invasive procedures allow a full weight-bearing after 1 to 2 weeks and the use of a cast is recommended for up to 6 weeks [15].

## 4. Discussion

AT is a rather frequent condition, based on the patient's social and sports involvement it can have a huge impact on the quality of life as described by Ceravolo et al. [85].

The early diagnosis and prevention of AT represent a real challenge since the early signs of AT are essentially attributable to the only manifestation of pain that limits sports participation. Although there are some signs and symptoms that may precede the onset of pain, such as morning stiffness and mild pain, these are often ignored by patients and doctors, making it even more difficult to implement prevention programs [1].

The lack of high-level guidelines makes AT management challenging. Diagnosis is generally based on the clinical examination and requires the physician's experience in recognizing areas of intratendinous swelling and tenderness of the tendon that disappears when tensioned. Silbernagel et al. [1] recommended the use of diagnostic tests such as pain on palpation, arch sign and RLHT to confirm AT, in fact as reported by the authors the recommendation is based on consistent and good quality evidence oriented to the patient.

The literature tends to agree in dealing with tendinopathy starting with conservative treatment by entrusting the patient to rehabilitation. The rehabilitation team made up of medical doctors, physiotherapists and athletic trainers should work in synergy, sharing a timeline and short, medium, and long-term goals for a complete recovery. The patient with AT should start an early rehabilitation program, under physiotherapist supervision, including eccentric exercises whose effectiveness has been demonstrated, indeed a recent systematic review confirmed that among the several conservative treatments of AT, the only one universally accepted as a gold standard is the eccentric exercise widely used in the clinical trials included in the review [86]. In addition to physical exercise, other conservative therapies can be used, such as the use of ESWT, physical therapies and the use of injections. It should be noted that these therapies alone are not enough to face up to AT, but they could help in addition to physical exercise. After at least 3–6 months of conservative treatments, surgical procedures could be suggested to the patient. Surgical management seems to be effective in 75–85% of cases, with a return to sports after 6 months.

This work certainly has limitations: this paper is not a systematic review therefore it is possible that not all relevant references have been included. Future works should focus

on systematically reviewing the literature to provide guidance for each of the chapters addressed in this overview.

**5. Conclusions**

Clinical examination is a key element in the diagnosis of AT, the RLHT showed moderate clinical evidence in the diagnosis and assessment of early symptoms is crucial for early initiation of conservative treatments.

Eccentric exercise represents the gold standard conservative treatment for AT. Regardless of the chosen therapeutic exercise modality (eccentric, combined, HSRT) it is important to follow a progression of loads based on the evaluation and clinical history of the patient.

Frequency of rehabilitation sessions must take into account the response times of the collagen tissue to load.

Surgery should be considered only when conservative management has failed; in post-surgical rehabilitation, the granting of weight-bearing must be carefully considered based on the type of surgery.

**Author Contributions:** Conceptualization, M.B. and F.S.; writing—original draft preparation, F.S., M.B.; writing—review and editing, M.B., F.S., F.B., S.M., S.S.; supervision, M.B., S.S. All authors have read and agreed to the published version of the manuscript.

**Funding:** The authors received no funding for this work.

**Institutional Review Board Statement:** Not applicable.

**Informed Consent Statement:** Not applicable.

**Conflicts of Interest:** The authors declare no conflict of interest.

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
