# Peer review of "An Overview of Achilles Tendinopathy Management"

_2673-4036, doi:10.3390/osteology1040017_

Round 1

Reviewer 1 Report

Comments and Suggestions for Authors

An overview of Achilles tendinopathy management

Thank you for giving me the opportunity to review for Osteology journal. This is very good review on Achilles Tendinopathy. Authors provide an organized review of the literature which includes diagnosis methods and several treatments. I comment the Authors for the effort. The manuscript is in a good shape and minor things need to be addressed. Bellow I provide my main concerns and suggestions. Again, I really enjoy reading this paper.

Abstract:

The abstract is well written and provide a good overall of the manuscript. However, there are a few things that need to change to provide more clarity. I suggest to authors to include inside the abstract the following answers: Why this review was performed? What are the research questions? What was the purpose of this review?

Consequently, background and “methods” inside the abstract needs to be restructured. Also, there are no methods inside the manuscript. Why Authors present methods inside the abstract?

Also, I suggest to Authors to use different key words not included in the title.

Introduction:

The intro is well written and provides a good narration of the background.

Please, read the instructions for Authors from Osteology journal and provide the in-text references in a common and specific way. Please, apply this comment throughout the manuscript.

Line 35: Capitalize the “the”.

Line 37: Please add a dot following the reference [8].

Lines 44-45: Question: Is there a possibility when the tendon is not well-prepared and receive a high-load high-intensity training program to be another reason for AT.

Although I am really pleased with the flow of introduction there is a lack of the research questions and the necessity of this review. Thus I suggest to Authors to add inside the last two paragraphs the research questions. Why this review was conducted? What are the main questions that need to be answered for readers? What is/are the main problem(s) that this review will solve here? This will provide a better understanding of the purpose of the review as well as a guide for the rest of the paper.

Diagnosis:

I enjoyed this part. Authors provide a crystal clear analysis of the diagnosis of AT. Although references need a common and standard writing everything was in a good shape. Images are really helpful here for the tests and the analysis is very good for the readers.

Line 110: Is there a scale of pain measurement for these two tests?

The only thing that missing here is a conclusion (short take home message) after each paragraph. For example, from the clinical measures of diagnosis what is suggested of being the best for patients? Αre there specific indicators that each patient should follow for each diagnosis? Similar for the imaging diagnosis methods.

Treatment:

This paragraph also provides a good analysis of treatments. Again, be consistent with the in-text references. Also, add a small take home message following each paragraph to summarize the main findings.  

Line 148: Change athletes to patients.

Line 159: Heel drops were performed with the injured leg only? Please clarify this here.

Line 161: Please delete the years of publication before or after the reference. Apply this throughout the manuscript.

Figure 2: The title of figure 2 is not in line with the text. Please clarify how perform this exercise. Also, would it help to have a small high step under the toes?

Extracorporeal shockwave therapy comes with pain. Although the duration of therapy is usually not exceeding the 2 minutes the majority of physiotherapists prepare patients that this therapy comes with pain. How deterrent can this work for patients?

Lines 220-224: Is this including heating ointments? Are there any research evidence for that?

Lines 225-227: Cryotherapy considers being the best treatment for tendonitis especially follows a high-load and high-intensity training program. Please, provide some more details here.

Discussion:

Although discussion is small, it provides a good presentation of the findings. I suggest to Authors to add some of the conclusions that will be drawn from the text above.

Line 285-286: I am losing the sentence here. I believe that “it” has to change to “and”.

Lines 296-298: These lines are super. Please highlight this suggestion for readers.

Conclusions:

I suggest to Authors to add bullets the main findings or suggestions from this review according to the paragraphs presented. Provide the main take home messages.

Reviewer 2 Report

Comments and Suggestions for Authors

This study is in my opinion of relevance to medicine sciences and would fit the scope of the journal. However, the authors need to specify the type of the paper. Accordingly, the paper should be considerably restructured and all criteria for a particular type of the paper have to be fulfilled. It can not be published in the present form.

Reviewer 3 Report

Comments and Suggestions for Authors

Thank you for the time and effort you have put into your manuscript. I do believe that more work needs to be completed for this article.

Firstly, the type of manuscript is missing (line 1). It might be only formal mistake.

According to the type of the paper please edit the whole text structure according to the Instructions for authors at mdpi.com, are very helpful.

It is strongly encouraged to authors to use the style of structured abstracts, but without headings (see the Instructions for authors at mdpi.com). Stay focused.

In redesigned structure identify and create key paper sections: Introduction, Material and Methods, Discussion, Limitations, Conclusions.

Section Material and Methods is missing. What databases were scoped? What were inclusion and exclusion criteria for use of the literature sources?

Provide the limitations of the study.

Discussion needs to be make over. Please discuss more results from literature, pros and cons, and provide implementation for praxis. 

Conclusion: shortly summarize the key message of the paper.

The amount of references looks at first glance satisfactory, but the process of their selection is not provided (section Materials and Methods).

Some sentences are difficult to read and message is hardly to catch. Please pay attention to punctuation and grammar errors.

According to the flaws mentioned above, I do not recommend this paper for publishing.

Round 2

Reviewer 1 Report

Comments and Suggestions for Authors

No comments

Author Response

We thanks the reviewer.

Best regards

M. Bravi

Reviewer 3 Report

Comments and Suggestions for Authors

Thank you for the time and effort you have put into your first manuscript and its revision. I think a lot of work have been done to complete this article.

The article is a narrative review/overview with sufficient analysis of literature and pivotal papers.

The text structure is divided according to the instructions and it is clear and easy to read and understand.

Revised version of abstract is aimed and concise, newer the less the style is still structured.

Keyword according to recommendation should not include the same worlds as the title of article.

The limitations of study were added as requested.

At the end of chapter introduction paragraph Material and Methods was added according to recommendation. Paragraph indent is missing.

Chapter Discussion is easier to read and gives to reader sufficient overview and provides a good presentation of the findings.

Chapter Conclusion is revised and at the moment focused.

Minor changes:

Table 1 and table 2 are not listed in the body of the manuscript and are not explained.

Please, try to avoid paragraphs of one sentence.

Please, pay attention to punctuation and some grammar errors. F. e. Line 330 AT,the ; Line 331 AT.Assessment.

According to the flaws mentioned above, I recommend this paper for publishing with minor changes.

Author Response

We thank the reviewer for the comments:

Table 1 and table 2 are not listed in the body of the manuscript and are not explained.

Table 1 is listed in the body of the manuscript line 70 and Table 2 line 156. We modified the tables for a easier comprehension. 

Please, try to avoid paragraphs of one sentence.

Please, pay attention to punctuation and some grammar errors. F. e. Line 330 AT,the ; Line 331 AT.Assessment.

According to the flaws mentioned above, I recommend this paper for publishing with minor changes.

Thanks for the comments, we revised the paper and corrected as suggested.